

# Biofilm released cells can easily be obtained in a fed-batch system using *ica+* but not with *ica-* isolates

Vânia Gaio and Nuno Cerca

Centre of Biological Engineering (CEB), Laboratory of Research in Biofilms Rosário Oliveira (LIBRO), University of Minho, Braga, Portugal

## ABSTRACT

*Staphylococcus epidermidis* is one of the major opportunistic bacterial pathogens in healthcare facilities, mainly due to its strong ability to form biofilms in the surface of indwelling medical devices. To study biofilms under in vitro conditions, both fed-batch and flow systems are widely used, with the first being the most frequent due to their low cost and ease of use.

**Aim**. To assess if a fed-batch system previously developed to obtain biofilm released cells (Brc) from strong biofilm producing *S. epidermidis* isolates could also be used to obtain and characterize Brc from isolates with lower abilities to form biofilms.

**Methodology**. The applicability of a fed-batch system to obtain Brc from biofilms of 3 $ica^+$ and 3 $ica^-$ isolates was assessed by quantifying the biofilm and Brc biomass by optical density (OD) and colony-forming units (CFU) measurements. The effect of media replacement procedures of fed-batch systems on the amount of biofilm was determined by quantifying the biofilm and biofilm bulk fluid, by CFU, after consecutive washing steps.

**Results**. The fed-batch model was appropriate to obtain Brc from $ica^+$ isolates, that presented a greater ability to form biofilms and release cells. However, the same was not true for $ica^-$ isolates, mainly because the washing procedure would physically remove a significant number of cells from the biofilm.

**Conclusions**. This study demonstrates that a fed-batch system is only feasible to be used to obtain Brc from *S. epidermidis* when studying strong and cohesive biofilm-forming isolates.

# INTRODUCTION

*Staphylococcus epidermidis* is a well-known nosocomial pathogenic associated with recurrent biofilm-infections, acknowledged as the major agent involved in biofilm-associated medical devices infections (*Becker, Heilmann & Peters, 2014*). Importantly, this bacterium, which was previously seen as a commensal microorganism due to its benign relationship with the host (*Cogen, Nizet & Gallo, 2008*; *Gardiner et al., 2017*), is nowadays accepted as an important opportunistic pathogen, of particular concern in ill and immunocompromised patients (*Otto, 2009*). *S. epidermidis* infections are more likely to happen upon invasive procedures involving indwelling medical devices, in which the

Corresponding author
Nuno Cerca,
nunocerca@ceb.uminho.pt

physiological barriers are compromised, since this bacterium is a ubiquitous inhabitant of the skin and mucosae in humans (*Ziebuhr et al., 2006*) and has a strong ability to form biofilms on the surface of medical devices (*Cerca et al., 2005c*; *Laverty, Gorman & Gilmore, 2013*). Bacteria within biofilms are undoubtedly more resistant to antibiotics (*Albano et al., 2019*; *Cerca et al., 2005a*; *Dias et al., 2018*) and to the host immune defense (*Cerca et al., 2006*; *Gray et al., 1984*; *Yao, Sturdevant & Otto, 2005*), contributing to the persistence and recurrence of infections (*Mah, 2012*; *Schommer et al., 2011*; *Singh & Ray, 2014*). For all these reasons, biofilms have been a major research target and extensive studies allowed to characterize the biofilm lifecycle and divide it into three main stages: attachment, maturation and disassembly (as reviewed in *Boles & Horswill, 2011*; *Otto, 2013*). The importance of a better characterization of the disassembly process in biofilms has been pointed out, since cells released from the biofilm can enter the systemic circulation and contribute to the spreading of the infection (*Boles & Horswill, 2011*; *Kaplan, 2010*) and cause severe systemic diseases, as bacteraemia (*Cervera et al., 2009*; *Wang et al., 2011*) which are associated with high levels of morbidity and mortality among immunocompromised patients (*Kleinschmidt et al., 2015*; *Rogers, Fey & Rupp, 2009*).

Both fed-batch and dynamic systems have been used to study and characterize initial adhesion (*Cerca et al., 2005b*; *Isberg & Barnes, 2002*) and maturation of the biofilm (*Moormeier & Bayles, 2014*; *Periasamy et al., 2012*). However, both present advantages and drawbacks, depending on the main focus of the study (*Bahamondez-Canas, Heersema & Smyth, 2019*). The few studies addressing disassembly rely almost entirely on dynamic systems, which is not surprising, as these systems present key advantages such as a controlled flow, allowing a continuous diffusion of oxygen, nutrients and waste, and are thought to be a more accurate representation of the conditions in which biofilms are formed in various diseases, as previously reviewed (*Azeredo et al., 2017*; *Bahamondez-Canas, Heersema & Smyth, 2019*). However, these systems are significantly more expensive and are often more difficult to assemble, being essential to have good background knowledge on hydrodynamics to study biofilms in such conditions (*Yawata et al., 2016*). Hence, it is no wonder that fed-batch systems are more frequently used on biofilm research, since they are easier to implement and already widely used under in vitro conditions (*Azeredo et al., 2017*; *Bahamondez-Canas, Heersema & Smyth, 2019*). Thus, the ability to implement fed-batch systems to high-throughput research in biofilms disassembly would be beneficial, as it would allow more studies to be undertaken on this research topic.

Earlier, we demonstrated the feasibility to use a fed-batch system to obtain *S. epidermidis* cells released from biofilms (Brc) (*França et al., 2016a*; *Gaio & Cerca, 2019*). However, we failed to include low biofilm-forming isolates on those studies and, as a consequence, the applicability of this model on such isolates could be questioned. Hence, the aim of the current study was to better understand the limitations of a fed-batch system to obtain Brc from *S. epidermidis* biofilms, by testing its potential to characterize Brc from $ica^+$ and $ica^-$ isolates with distinct abilities to form biofilms.

## MATERIALS & METHODS

### Bacterial isolates and growth conditions

Six isolates of *S. epidermidis*, with different abilities to form biofilms and characterized by the presence (+) or absence (-) of the intercellular adhesion gene (*ica*), generally involved in *S. epidermidis* biofilm formation (*Cafiso et al., 2004*) were selected to conduct this study (Table 1). Growth conditions followed the fed-batch model previously described to obtain Brc from *S. epidermidis* (*França et al., 2016a*). First, a colony of *S. epidermidis* was inoculated into two mL of Tryptic Soy Broth (TSB) (Liofilchem, Teramo, Italy) and incubated overnight at 37 °C with shaking at 120 rpm in an orbital shaker. The overnight inoculum was then diluted in the same growth medium to reach an optical density (OD) of $0.250 \pm 0.05$, measured at 640 nm, which corresponds to a concentration of approximately $2 \times 10^8$ CFU/mL (*Freitas et al., 2014*). To form biofilms, 15 µL of the previously adjusted suspension were added to one mL of TSB supplemented with 0.4% (v/v) glucose (TSBG) to induce biofilm formation, into a 24-well microtiter plate (Orange Scientific, Braine-l'Alleud, Belgium), that was incubated in an orbital incubator at 37 °C with agitation at 120 rpm, for as long as 72 ($\pm$1) hours. Spent medium was carefully removed after each 24 ($\pm$1) hours of incubation, followed by washing twice the biofilm with a 0.9% (m/v) NaCl solution to remove unattached cells. Next, one mL of fresh TSBG was carefully added to the biofilms and the plate was incubated in the same temperature and agitation conditions. Then, at either 28, 48 or 72-hours of growth, the supernatant was removed and biofilms were washed twice with saline solution. Remaining biofilms cells were scraped from the microtiter plate with the aid of a plastic tip and resuspended in one mL of the NaCl solution. Cells were pooled together from at least 4 distinct disrupted biofilms to decrease biofilm formation variability (*Sousa, França & Cerca, 2014*). Planktonic cultures were grown in an orbital shaker for 24 ($\pm$1) hours at 37 °C with shaking at 120 rpm. Finally, Brc were carefully aspirated from the biofilm bulk fluid after 28, 48 and 72 h of growth. A schematic representation of the method used to culture and collect the populations mentioned in this section was included in Data S1.

### Homogenization and quantification of the populations

Before quantification, all 3 populations (disrupted biofilm cells, Brc and planktonic cells) were homogenized by sonication through a pulse of 5 s at 40% amplitude (Ultrasonic Processor Model CP-750, Cole-Parmer, Illinois, USA). As shown before, this sonication cycle did not affect cell viability (*Cerca et al., 2005a*). The total biomass of all bacterial populations was quantified by OD measurement at 640 nm ($OD_{640}$), as previously optimized (*Freitas et al., 2014*). At least three independent experiments, with technical duplicates, were performed.

### Effect of consecutive biofilm washing on cell detachment from the biofilms

Biofilms were formed for 24 h, as described above. Then, the supernatant was carefully removed and the total number of cells on the supernatant was quantified by CFU. Biofilms were then washed with a saline solution, up to 6 consecutive times. Between each wash,

**Table 1  Origin of the 6 *Staphylococcus epidermidis* isolates used in this study.**

| S. epidermidis isolate | Description | Country of origin | Ica operon |
| --- | --- | --- | --- |
| 9142 (*Mack, Siemssen & Laufs, 1992*) | Clinical isolate from blood culture | Germany | Detected (*Cerca et al., 2013*) |
| DEN69 (*Cerca et al., 2013*) | Unknown | Denmark | Detected (*Cerca et al., 2013*) |
| PT13032[a] | Clinical isolate from a patient with chronic renal failure | Portugal | Detected[a] |
| ICE102 (*Cerca et al., 2013*) | Clinical isolate from a patient with infective endocarditis | Iceland | Undetected (*Cerca et al., 2013*) |
| DEN185 (*Cerca et al., 2013*) | Unknown | Denmark | Undetected (*Cerca et al., 2013*) |
| PT12004 (*Freitas et al., 2017*) | Clinical isolate from a patient with chronic renal failure | Portugal | Undetected (*Freitas et al., 2017*) |

Notes.

[a]Unpublished isolate obtained from a previous epidemiological study in Portugal. Isolates were obtained after patient informed consent with the approval of the Ethical Committee of the Hospital Geral de Santo Antnio, Porto, Portugal. Each isolate was first identified at the species level using the commercially available VITEK® two identification system using the gram-positive ID card (BioMérieux, Marcy lÉtoile, France) and molecular identification was confirmed by sequencing of the *rpoB* gene (*Mellmann et al., 2006*).

bacteria in the supernatant were quantified by CFU. Simultaneously, the quantification of CFU of the remaining biofilm was done after 1, 2 and 6 washes. Four independent assays were performed for each strain and technical duplicates were used.

## Quantification of active dispersion of cells from 24 h biofilms

After discarding the spent medium and washing twice the preformed 24 h biofilms, one mL of TSB was carefully added to the wells. In half of the biofilm wells, the newly added TSB was immediately transferred into empty sterile wells, as described in Fig. 1. This medium contained cells released from the pre-established biofilm (Brc), due to the shear forces exerted by medium addition, as determined before (*França et al., 2016a*). The plates were incubated at 37 °C with shaking at 120 rpm. At different time points, a 20 µL aliquot was collected from both conditions. The number of cultivable cells was determined by CFU. Four independent experiments with three technical replicates were performed.

## Comparison of the antibiotic susceptibility of Brc collected at distinct time points

Brc were collected after 28 h, 48 h and 72 h of biofilm formation. The 28 h time point was included to assess the effect of Brc physiology 4 h after the first medium removal. The bacterial cell concentration was adjusted by OD to a final concentration of around $2\times 10^8$ CFU/mL and bacterial suspensions were incubated with peak serum concentrations (PSC) of vancomycin (40 mg/L), rifampicin (10 mg/L) or tetracycline (16 mg/L) (*National Committee for Clinical Laboratory Standards, 1997*) for 2 h at 37 °C with agitation at 120 rpm. Controls were performed in simultaneous by incubating the suspensions in the same conditions, without the addition of the antibiotics. The effect of the antibiotics was assessed by CFU counting upon 10-fold serial dilutions and plating into Tryptic Soy Agar (TSA) plates. This assay was performed with technical duplicates and at least three independent times.

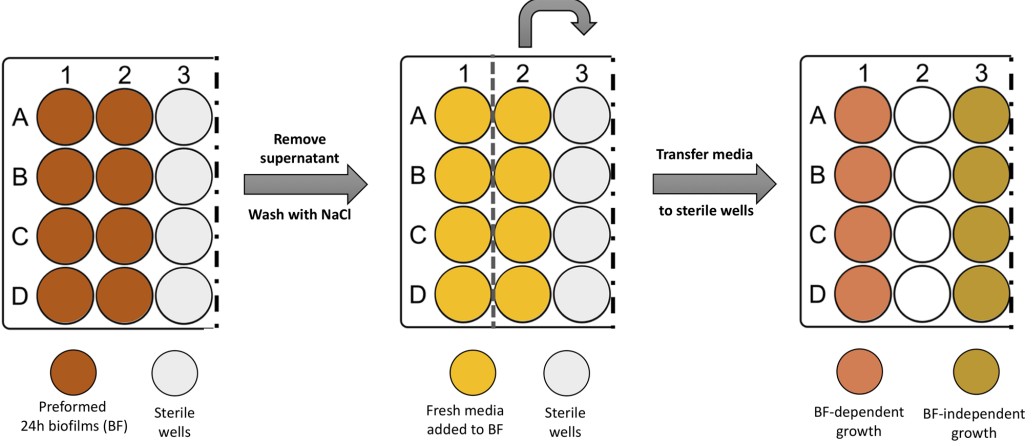

**Figure 1** Scheme illustrating the process of obtaining biofilm released cells using a fed-batch growth system.

## Statistical analysis

Statistical significance between consecutive washes performed on biofilms (Fig. 2) and between control and antibiotic-treated samples (Data S2) was determined with one-way ANOVA multiple comparisons ($p < 0.05$). Statistical difference regarding growth kinetics in the presence or absence of the originating biofilm (Fig. 3) was determined using multiple $T$-tests ($p < 0.01$). All analysis was performed using GraphPad Prism version 6 (Trial version, CA, USA). At least three replicates (independent experiments) were performed for all assays.

## RESULTS

### Characterization of biofilm formation and Brc collection by multiple *S. epidermidis* isolates

A total of 6 distinct *S. epidermidis* isolates, previously characterized regarding their biofilm formation ability and the presence of biofilm-associated genes, namely the *ica* operon, were selected for this study (*Cerca et al., 2013*; *Freitas et al., 2017*). Initially, biofilms were grown up to 72 h. The ability to produce Brc over time, using the implemented fed-batch system, was evaluated by calculating the ratio between the number of cells existing in the biofilm bulk fluid and within the biofilm biomass (Bbf/B) (Table 2). From the 6 isolates used in this study, all 3 $ica^+$ isolates produced remarkably more biofilm than isolates without the *ica* gene, especially after 72 h of incubation. This was not surprising since several studies showed a relation between the presence of the *ica* locus and the increased ability to form biofilms (*Heilmann et al., 1996*; *Mack et al., 1994*; *Qin et al., 2007a*), despite also being known that some $ica^-$ isolates are also able to produce biofilms (*Dice et al., 2009*; *O'Gara, 2007*; *Qin et al., 2007b*; *Tormo, 2005*). Interestingly, it was observed that the thickest biofilms produced had a lower Bbf/B ratio.

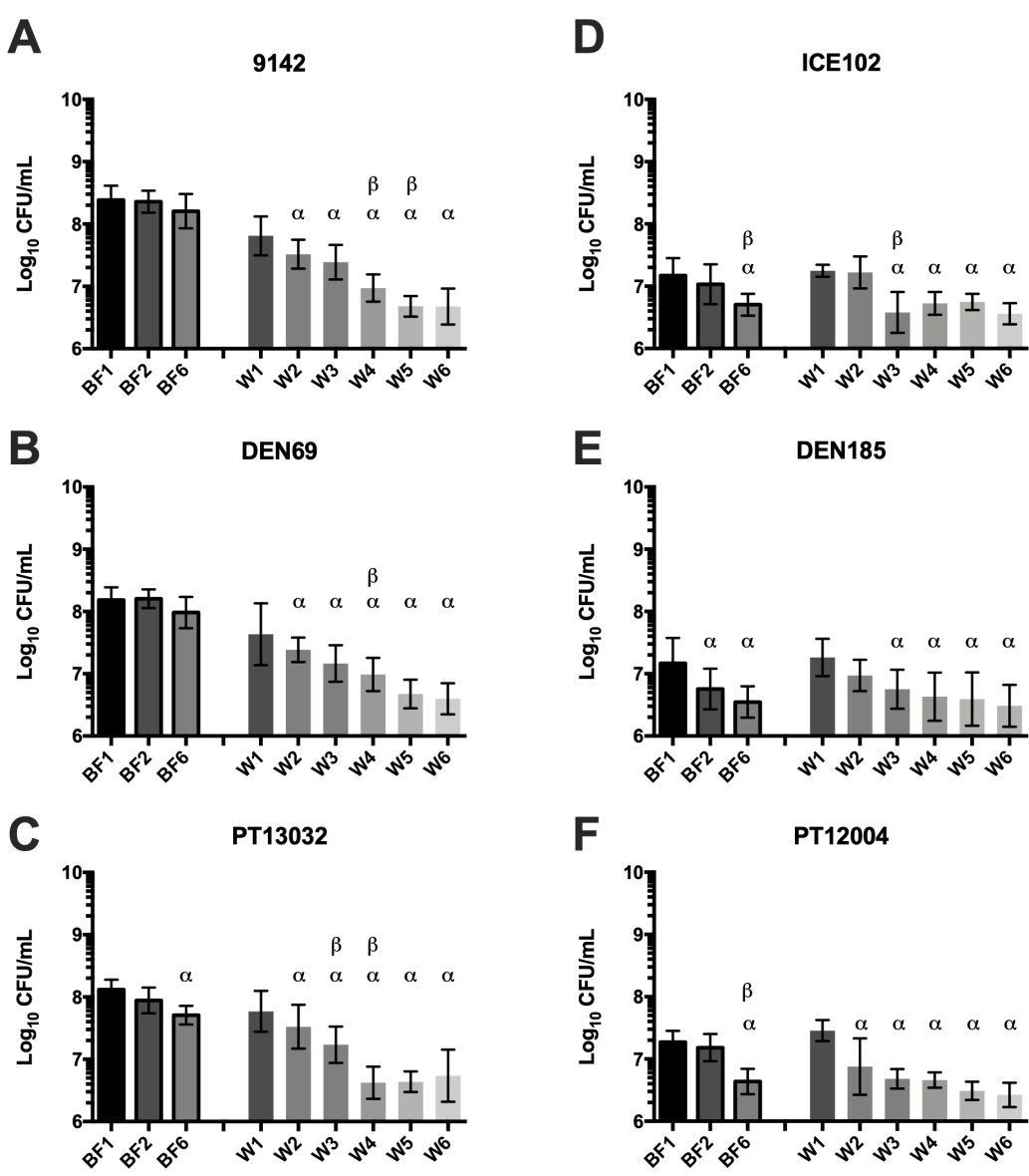

**Figure 2** Log₁₀ number of CFU in the biofilm or in the wash liquid after continuous washing steps in (A–C) *S. epidermidis ica⁺* isolates or (D–F) *S. epidermidis ica⁻* isolates. CFU were determined from biofilms after 1 (BF1), 2 (BF2) or 6 washes (BF6) and from the NaCl solution obtained after each washing step (W1 to W6). The columns represent the mean plus or minus standard error deviation of at least three independent experiments, with duplicates. Statistical differences were analyzed with one-way ANOVA multiple comparisons, with α representing statistically significant differences ($p < 0.05$) between the first condition (BF1 or W1) and all remaining conditions, while β represents significant differences ($p < 0.05$) between each consecutive washing step.

## Washing biofilm and replacing the growth medium in fed-batch systems triggers the physical detachment of biofilm cells

It was previously shown that the typical medium replacement procedures needed for fed-batch systems trigger the detachment of cells from the biofilm due to shear forces

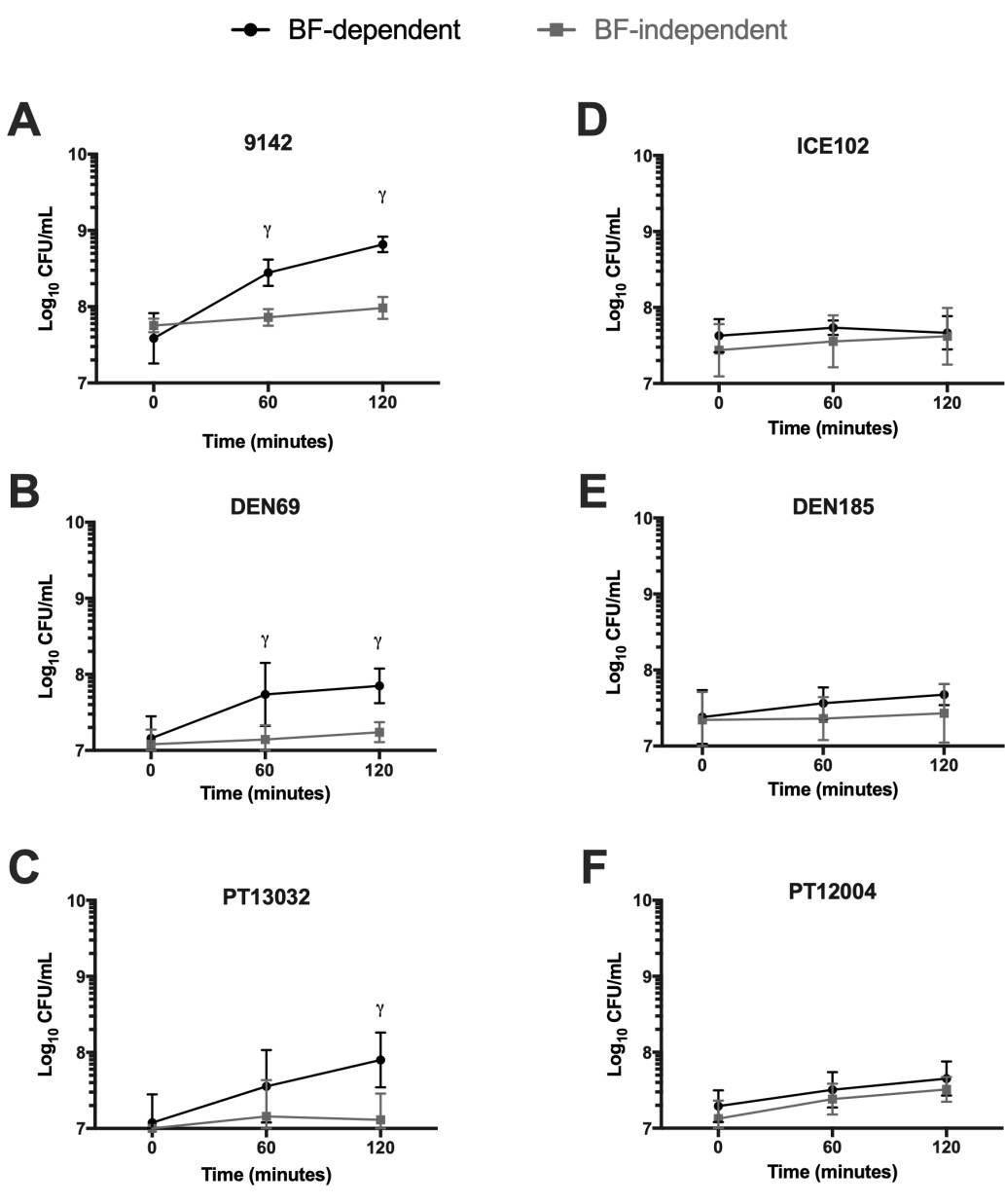

**Figure 3** Growth kinetics ($Log_{10}$ CFU/mL) of Brc in the presence or absence of the originating biofilms, under agitation (120 rpm) in (A–C) *S. epidermidis ica*[+] isolates or (D–F) *S. epidermidis ica*[−] isolates. The number of CFU was determined at 0, 60 and 120 minutes after media change in the presence (BF-dependent) or absence (BF-independent) of the originating biofilm. The columns represent the mean plus or minus standard error deviation of at least three independent experiments. Statistical differences were analyzed with multiple *T*-tests, with $\gamma$ ($p < 0.01$) representing statistically significant differences between the number of cells recovered at each time point between growth in both conditions (BF- dependent and -independent).

(*França et al., 2016a*). Using the selected isolates, a total of 6 consecutive washing steps were performed on each biofilm, followed by quantification of the number of cells released immediately after each wash, as well as cells remaining in the biofilm (Fig. 2). Interestingly,

**Table 2** Characterization of biofilm biomass (by OD measurements) and the ratios between the biofilm bulk fluid, containing Brc, and the biofilm suspension (Brc/Bio).

| Classification | *S. epidermidis* isolates | OD Biofilm | | | Ratio Bbf/B | |
|---|---|---|---|---|---|---|
| | | 28H | 48H | 72H | 28H | 48H |
| | **9142** | 1.65 | 2.54 | 3.05 | 0.64 | 0.47 |
| *ica*⁺ | **DEN69** | 1.92 | 1.92 | 2.26 | 0.51 | 0.79 |
| | **PT13032** | 0.48 | 0.88 | 1.29 | 1.10 | 1.29 |
| | **ICE102** | 0.61 | 0.81 | 0.89 | 2.12 | 3.65 |
| *ica*⁻ | **DEN185** | 0.43 | 0.78 | 0.77 | 2.85 | 3.63 |
| | **PT12004** | 0.57 | 0.69 | 0.72 | 1.81 | 2.77 |

all strains used each successive wash kept detaching cells from the biofilm, ranging from ~$10^7$ to ~$10^6$ CFU/mL per wash, independently of their biofilm formation capacity. Remarkably, the total biomass of stronger biofilms producers (9142, DEN69 and PT13032) was only moderately affected after the 6 washes, with circa 40% decrease of total biofilm biomass from the first to the last washing step. Conversely, *ica*⁻ isolates (ICE102, DEN185 and PT12004) biofilm structure was relatively more affected by shear forces, with more than 70% of the biofilm being removed after the 6 washes ($p = 0.03$).

## A higher number of cells is released to the supernatant when growing in the presence of strong biofilms

To differentiate between physical detachment and active dispersion of cells, a second experiment was performed. After washing a preformed 24 h biofilm and replacing the growth medium, the total number of cells in the bulk fluid was quantified right after medium replacement and after 2 h of incubation. In half of the wells, the bulk fluid containing Brc was transferred to new sterile wells. As shown before for strain 9142 (*França et al., 2016a*), the presence of the originating biofilm significantly increased the number of cells in the bulk fluid of *ica*⁺ strains, especially at 120 min (Fig. 3), when compared to the inoculum transferred to new sterile wells (in the absence of the preformed biofilm). Conversely, the same was not true for the *ica*⁻ isolates, since the effect of growth in the presence of the biofilm was significantly less pronounced and no statistical differences between growing in the presence or absence of the originating biofilm were found, growing up to 120 min. This is not surprising since we know from the previous experiment that a significant number of cells was removed from the weakest biofilms, leaving a low amount of cells available to be released.

## Brc obtained at different time points present the same antimicrobial susceptibility to vancomycin, tetracycline and rifampicin

Previously, we observed that Brc collected from 24 h preformed biofilm, up to 4 h after medium replacement, presented enhanced tolerance to vancomycin, tetracycline and rifampicin (*França et al., 2016a*). As a complementary analysis to be included in this study, we decided to test if this enhanced antimicrobial tolerance was somewhat influenced if the cells were collected from more mature biofilms, namely from 48 and 72 h biofilms. To

assess this, the antimicrobial susceptibility of Brc obtained at 28, 48 and 72 h to peak serum concentrations of vancomycin, tetracycline and rifampicin was determined. Interestingly, no significant differences in the tolerance to vancomycin, tetracycline or rifampicin were found between the different Brc populations, obtained 4 h (28H biofilms) or 24 h (48H/72H biofilms) after medium replacement (Data S2). This data is indirect evidence suggesting that the overall phenomenon of Brc is not affected by the maturity of the biofilm.

## DISCUSSION

It has been pointed out that during biofilm disassembly phase, cells are released from the biofilm to the surrounding environment, spreading the infection and increasing the risk of systemic diseases, as bacteraemia (*Boles & Horswill, 2011*; *Cervera et al., 2009*). Recently, we demonstrated the feasibility of using a fed-batch system to obtain Brc from high biofilm-forming *S. epidermidis* strains (*França et al., 2016a*), and showed that Brc had enhanced tolerance to antibiotics (*Gaio & Cerca, 2019*) and also induced a more inflammatory response in the host (*França et al., 2016b*). An important limitation of the previous studies was the fact that we only tested *ica*$^+$ *S. epidermidis* isolates, with considerable abilities to form thick and multi-layered biofilms (*Christensen et al., 1985*; *Mack et al., 1994*; *Mack, Siemssen & Laufs, 1992*). As inter- and intra-species variability has been observed regarding *Staphylococcal spp.* biofilm formation (*Handke et al., 2004*; *Oliveira et al., 2015*; *Tremblay et al., 2013*), it was important to determine if the previous findings were reproducible when using strains with lower ability to form biofilms. Since it is well known that strains without a functional *ica* operon form biofilms with lower biomass (*Handke et al., 2004*), we compared 3 *ica*$^+$ and 3 *ica*$^-$ isolates.

Not surprisingly, significant differences were found between *ica*$^+$ and *ica*$^-$ isolates, regarding the effect of shear forces on the biofilm biomass reduction. A higher proportion of cells was found to be detached from weaker biofilms, as well as the decrease on total biofilm biomass was significantly more pronounced on *ica*$^-$ isolates, while a similar number of cells was being removed from the second to the last washing step, suggesting an almost inversely proportional ability to physically detach cells from the biofilms as related to the biofilm cohesiveness (*Mack et al., 1996*). The opposite was found for stronger biofilms, as it seems that a higher number of cells was recovered from initial washes (W1 and W2), but remarkably lower amounts of cells were detached in the last stages of washing, presumably because deeper layers of the biofilm are more cohesive and resistant to shear forces.

We also assessed if the enhanced antimicrobial tolerance described before (*França et al., 2016a*) was dependent on biofilm maturation stage. By obtaining Brc from ~1, 2 and 3 day-old biofilms, we were able to determine that the effect observed in early-stage biofilms also occurred in older biofilms.

As noted before, a key limitation of using a fed-batch model to originate Brc is the difficulty to differentiate between physically detached cells, resulting from the washing procedures, from actively dispersed cells (*Boles & Horswill, 2011*; *França et al., 2016a*; *Kaplan, 2010*). As shown with the multiple washing steps experiment, our data confirm that shear forces exerted during washing and medium replacement trigger the detachment

of cells, independently of the ability of the isolates to produce thicker or thinner biofilms or the number of washes involved. However, active dispersion could only be determined in the $ica^+$ strains tested. Conversely, for the strains without a functional $ica$ operon, the quantity of cells on the biofilm bulk fluid incubated in the presence or absence of the biofilm was generally the same. This phenomenon in $ica^-$ isolates may be a consequence of the large proportion of cells removed upon washing their weak biofilms, which led to a higher proportion of cells in the supernatant immediately upon the addition of fresh media and, consequently, to a lower availability of cells in the biofilm to be continuously released. On the other hand, stronger biofilms were less affected by the washing steps used to remove non-adherent cells, leading to a lower proportion of cells detached from shear forces, compared to the originating biofilm, and, consequently, to a higher concentration of cells actively released from the biofilm to the supernatant.

## CONCLUSIONS

The results obtained herein demonstrated that a fed-batch system is only reliable in obtaining Brc from *S. epidermidis* biofilms for some isolates, especially from those who can form thick and strong biofilms. While all $ica^+$ isolates used herein were found to be high biofilm producing strains, it should be noted that some $ica^+$ isolates lack a functional intact operon (*Cafiso et al., 2004*; *Cue, Lei & Lee, 2012*), and the mere presence of the gene might not be related to its expression (*Freitas et al., 2017*; *Lerch et al., 2019*). As such, to assess the feasibility of this method in more strains, it is important not only to determine the presence of $ica$ but to assess if the operon is functional, as mutations in major biofilm regulators may influence the dynamics of Brc production (*Cue, Lei & Lee, 2012*).

### Funding

This study was supported by the Portuguese Foundation for Science and Technology (FCT) under the scope of the strategic funding of UIDB/04469/2020 and by BioTecNorte operation (NORTE-01-0145-FEDER-000004) funded by European Regional Development Fund under the scope of Norte2020 - Programa Operacional Regional do Norte. The funders had no role in study design, data collection and analysis, decision to publish, or preparation of the manuscript.

### Grant Disclosures

The following grant information was disclosed by the authors:
Portuguese Foundation for Science and Technology (FCT): UIDB/04469/2020.
BioTecNorte operation: NORTE-01-0145-FEDER-000004.
Norte2020 - Programa Operacional Regional do Norte.

### Competing Interests

The authors declare there are no competing interests.

## Author Contributions

- Vânia Gaio performed the experiments, analyzed the data, prepared figures and/or tables, authored or reviewed drafts of the paper, and approved the final draft.
- Nuno Cerca conceived and designed the experiments, analyzed the data, authored or reviewed drafts of the paper, and approved the final draft.

## Data Availability

The raw data used to create Table 2, Figs. 2 and 3 are available in the Supplementary Files.

## Supplemental Information

Supplemental information for this article can be found online at http://dx.doi.org/10.7717/peerj.9549#supplemental-information.

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
