# Peer review of "Biofilm released cells can easily be obtained in a fed-batch system using ica+ but not with ica- isolates"

_PeerJ, doi:10.7717/peerj.9549_

## Round 0.1 · original submission · Minor Revisions

Congratulations on a well constructed manuscript. The reviewers have a few minor revisions for you. In addition, a couple of minor points from myself:

Line 130: do you mean scraped? And could you state what these were scraped with?

Table 1 - are the numbers here supposed to be references (e.g. 36 for ICE102)? Just the reference list is not numbered.
Is ICE102 definitely from the United States? Assuming this strain is from the Herminia de Lencastre's collection (I seem to recognise some of the strain numbers), it should be from Iceland (Table 1: https://www.ncbi.nlm.nih.gov/pmc/articles/PMC2768820/). Apologies if this is a different strain.

·

Basic reporting

no comment

Experimental design

no comment

Validity of the findings

no comment

Additional comments

Some errors found in file in red font so re-send the revised file to the journal

Reviewer 2 ·

Basic reporting

No comment

Experimental design

No comment

Validity of the findings

No comment

Additional comments

Comments

1.Is there any approach to conduct the experiment under static condition in addition to shaking condition culture? If so what was difference?
2.Is it 24h or 28h?

Overall the authors made nice attempt to characterize Brc from ica+ and ica- isolates with distinct abilities to form biofilms.

Reviewer 3 ·

Basic reporting

In this study, Gaio and Cerca address an open question, which arose from their previous study on the feasibility of a fed-batch system to obtain S. epidermidis cells release from biofilm from strains with lower abilities to form biofilms. I have a positive feeling about the publication as the study is well written, introduction is clear with the relevant literature being referenced, data are well presented and conclusions are appropriately stated

Experimental design

I have few suggestions/questions, which might help improving the manuscript:

Major points:
1) The fed-batch method is the focus of the manuscript; it includes various steps and different measurement at different times (described in lines 116-135). I think including a general scheme, which summarize this method (with the measurements, the naming of the samplings….), would be helpful for the reader as it is hard to keep track of the steps. For example:
- does the Planktonic culture in line 133 refer to the “supernatant” from line 129?
- To what do planktonic cells from line 138 refer?

The authors already have Figure 1, which summarized part of the method but not the whole.

2) Table 2. I am not sure I understand the ration Brc/Bio. This ratio is very high in the ica- isolates because these strains form no biofilm and the few cells just live has planktonic cells. Therefore, I would not call these cells “biofilm released cells” as there was no biofilm in the first place. These are cells, which detached from the almost not formed biofilm (as the authors also observed in the following experiment). Maybe the authors can just rephrase, and make it clearer?

Minor points:

Line 31: “To form biofilms ON the surface” and not ““To form biofilms IN the surface”
Line 78: “for all THESE reasons” and not “for all THE reasons”
Table2: Why did the authors not measure the Ration Brc/bio also on 72 hours?
Table 1: line 5, rpoB has to be written in italics
Figure 2 legend. “with α representing statistically significant differences” and not “with a representing statistically significant differences”

Validity of the findings

no comment

Additional comments

no comment

---

## Round 0.2 · accepted · Accept

Both reviewers felt that you had adequately addressed their previous concerns, and returned no additional feedback, so it is my pleasure to be able to accept your paper for publication.

·

Basic reporting

no comment

Experimental design

no comment

Validity of the findings

no comment

Additional comments

no comment

Reviewer 3 ·

Basic reporting

no comment

Experimental design

no comment

Validity of the findings

no comment

Additional comments

The authors nicely answer all the questions/comments arose from the reviewers